# Development of an Embedded Software and Control Kit to Be Used in Soilless Agriculture Production Systems

**DOI:** 10.3390/s23073706

**Published:** 2023-04-03

**Authors:** Uğur Yegül

**Affiliations:** Department of Agricultural Machinery and Technologies Engineering, Faculty of Agriculture, Ankara University, 06135 Ankara, Turkey; yegul@ankara.edu.tr; Tel.: +90-541-258-1983

**Keywords:** vertical farming, automation, hydroponics, agriculture, sensor, water quality

## Abstract

Traditional agricultural methods, which are still adopted today, depend on many factors. Food production processes face serious risks, such as limited clean water resources and supplements such as phosphorus, in addition to weather conditions due to climate change, the distribution of pests and soil-borne diseases, and increasing demand due to population growth, which can lead to famine. In particular, there may be cases where the soil nutrient content is insufficient and the soil structure is not suitable for cultivation. Therefore, soilless farming techniques have become popular, where the producer is entirely in control of the process. Additionally, many factors affect conventional farming techniques, including restrictions on land suitable for agriculture, climate-increased transportation costs from production areas to central regions, and environmental sanctions. Therefore, soilless farming techniques and the use of technology have rapidly gained importance. The use of technology has two crucial parameters: hardware and software. Today, no device can simultaneously control the electrical conductivity, pH, dissolved oxygen, and temperature of the solution in systems cultivated with soilless farming techniques. The present study was conducted to find a solution to the needs in this area. An automatic control system was developed and tested, employing a microcontroller, various sensors, appropriate open-source codes, and original software. Electrical conductivity (EC), power of hydrogen (pH), dissolved oxygen (DO), and temperature (T) values were determined successfully by the developed control system. The area where the experiment was conducted is a fully controlled and closed area established within Ankara University. The ambient temperature was 22 °C and the humidity was 39%. The coordinates of the experimental area are 39.962013 and 32.867491. Three different artificial lighting intensities (165.6 µmol m^−2^ s^−1^, 248.4 µmol m^−2^ s^−1^, and 331.2 µmol m^−2^ s^−1^) and a desired photoperiod duration can be applied to the site.

## 1. Introduction

Soilless agricultural production systems are environmentally friendly due to controlling inputs such as water, chemicals, and fertilizers. Additionally, energy is used with optimum efficiency in soilless agriculture. It offers much cleaner and more hygienic products compared to soil-based agriculture. The technology required for applying soilless farming techniques has developed rapidly, becoming widespread and gaining significant importance in recent years. This technology applies to methods in which all agricultural parameters can be controlled, and the area and water requirements can be used at a minimal level. Traditional farming depends on many parameters. Conditions such as soil nutrients, climatic conditions, and unsuitable soil structures for cultivation can adversely affect conventional agriculture. Furthermore, pests and diseases in the soil can significantly reduce overall production. Therefore, agricultural techniques in a controlled environment have gained significant importance recently. The most important type is “soilless agriculture”, which has become popular with changing climatic conditions and decreasing agricultural land [1].

Approximately 12% (1.5 billion hectares) of the world’s land area is used to produce agricultural products [2]. According to FAO’s 2030/2050 forecast report, the amount of arable land per capita for developing countries, developed countries, and the world generally will decrease yearly. While the world’s population and the demand for food are constantly increasing, the decrease in the average area of arable land per capita every year shows the probability of countries facing food crises in the coming years [2]. Additionally, although the area of arable land in the world is decreasing each day due to climate change and the misuse of agricultural land for non-agricultural purposes, the world’s population has been increasing faster than ever in recent years. While it took 20 years for the world’s population to increase by 100 million people 100 years ago, it increased by 83.3 million in 2018 alone [2]. In the face of this population increase, which threatens food security, countries have sought various solutions. As of today, cross-border agricultural investment and soilless farming methods have become necessary as a solution. In addition to the decrease in arable land area, the decrease in and pollution of freshwater resources are other critical problems in agriculture.

Especially since the beginning of 2020, with the emergence of COVID-19 in China, the pandemic declared by the World Health Organization has made trade wars based on food products more critical [3]. Under these conditions, where health problems and access to healthy food are important issues, closed systems that use water efficiently and save space have rapidly gained importance. Even people not interested in any agricultural activity have grasped the importance of organic agriculture and sought to cultivate their products.

Various pieces of hardware and software are needed to implement soilless farming techniques. One of these pieces of hardware is the Arduino microcontroller board, which can be considered cheap and easy to use. They can be easily connected to a computer via a Universal Serial Bus (USB) cable powered by an adapter or battery [4]. Furthermore, Arduino microcontroller boards can be programmed through a computer called a workstation in order to develop software [5].

Arduino is an electronic platform powered by open-source code based on hardware and software. It can read a value from a sensor, activate a motor, light an LED, transmit information to the Internet and a mobile phone, receive information, and convert these data into applications. The targeted action can be coded by sending instructions to the microcontroller on the board. Thanks to its simple and accessible user experience, Arduino has been used in thousands of different projects and applications. Arduino software is easy for beginners and flexible enough for advanced users. It works on Mac OS, Windows, and Linux, and the most crucial advantage is that it is inexpensive compared with other microcontroller platforms. Arduino software can be extended by experienced programmers via C++ libraries and can also be developed by circuit designers [6].

It is applied in different methods such as soilless agriculture or controlled farming, hydroponics, aggregate culture, and aeroponics. The plant’s growing conditions can be optimally controlled, especially the EC, pH, DO, and T values of the liquid in these production systems [7,8]. Electrical conductivity (EC) is the electrical resistivity of water, and the SI unit symbol Siemens (S) is the unit of electric conductance (mS cm^−1^). The power of hydrogen or potential of hydrogen (pH) measures the acidity or alkalinity of water-soluble substances: seven is neutral, lower values are more acidic, and alkaline and have no units. Dissolved oxygen (DO) is the amount of oxygen in water; the SI unit symbol is mg L^−1^. Temperature (T) represents the water temperature; the SI unit symbol is Kelvin (K) or Celsius (°C). Additionally, it is essential to minimize production risks regardless of soil structure, physics, and chemistry. For example, in soilless agricultural systems, cation exchange capacity, water buffering capacity, electrical conductivity, etc., are in the root zone of plants. Therefore, optimum physical conditions can be provided. With the nutrient solution, nutrition can be met at the optimum level in different plant growth periods and in states of changing temperature, light, humidity, and carbon dioxide/oxygen (CO_2_/O_2_) gas ratio [9].

Today, automation applications in agriculture and soilless agriculture have become widespread. The present study aimed to fill in some of the gaps encountered in these applications. While automation applications control specific parameters, IoT applications aim to determine environmental factors and exchange data. In modern agriculture, IoT technologies are often used to collect real-time information such as the temperature, humidity, wind, weather, rainfall, soil moisture, soil electrical conductivity, pH value, and soil nitrogen value. Smart agriculture is defined as the interactive operation of many technologies for agricultural purposes [10,11,12]. With the developing technology, smart devices, mobile phones, and thousands of devices connected to the internet have been developed. All technological or electronic devices (objects) produced are now addressable and usable in the network environment. IoT is a set of systems that control or regulate physical objects to see, perceive data, think, make decisions, share data, and communicate with each other. With IoT, the workload in innovative agriculture technologies can be reduced while at the same time increasing the yield and quality of the product. Many operations can be realized in IoT-based agricultural applications. Guo and Zhong applied IoT techniques to precise irrigation, fertilization, and air conditioning systems for more efficient greenhouse production [13]. Srbinaovska et al. proposed a wireless sensor network architecture for vegetable production in greenhouses and reduced management costs by monitoring environmental data [14]. These researchers designed a practical and cost-effective data monitoring system based on wireless sensor network technology to monitor basic environmental parameters such as temperature, humidity, and light. Türker et al. developed a prototype system to measure, wirelessly transmit, and monitor the temperature changes inside a greenhouse using the IoT method. The system consists of a Raspberry Pi, a Wi-Fi adapter, and two DS18B20 temperature sensors. The system records air, soil, and water temperature data in a greenhouse and transmits them to a database on the web [15]. Vincentdo and Surantha studied the control of pH and plant nutrients in soilless agriculture [16]. The present study aimed to provide the instant control of four different parameters and transfer data to the user. By examining the previous studies, it was observed that there are studies in which only pH control or only plant nutrient control, only pH and plant nutrient control, or only temperature and pH control were used, whereas short message services were not. No device can instantly control all of these parameters and transmit information to the user. 

Automation kits used in soilless farming systems are high-cost test devices. Therefore, the control kit obtained within the project’s scope will be fast and easy to use, allowing it to become widespread due to its affordable price and easy use. Therefore, a control kit was developed with an Arduino microcontroller, four different sensors, auxiliary equipment, and software prepared by the project coordinator. Although such controller cards are used in many sectors today, their use in agriculture is not at the desired level. Furthermore, today, no device can simultaneously control the electrical conductivity, pH, dissolved oxygen, and temperature of the solution in systems cultivated with the soilless farming technique. The present study was conducted to find a solution to the needs in this area.

## 2. Materials and Methods

The control kit included the following equipment: an Arduino microcontroller, a GSM module, a relay circuit, an EC sensor, a pH sensor, a DO sensor, a temperature sensor, four different submersible pumps to control EC and pH values, a submersible pump to control the dissolved oxygen amount, and a heater to maintain temperature values.

The functions of the developed kit include data collection, decision-making, and implementation. Additionally, the kit developed as a result of the study can be integrated into any hydroponic farming system of any size and control the EC, pH, DO, and T values of the liquid in the system thanks to the software developed with the microcontroller and various sensors.

The developed kit consists of the following equipment: an Arduino Uno R3 Board and a microcontroller board based on ATmega328P. It has 14 digital input/output pins (six can be used as PWM outputs), 6 analog inputs, a 16 MHz ceramic resonator (CSTCE16M0V53-R0), a USB connection, a power jack, an ICSP header, and a reset button. It is connected to a computer via a USB cable or powered with an AC-to-DC adapter or battery. The task of the Arduino microcontroller in the developed model is to work as the system’s brain, reading the data from the connected sensors to evaluate and communicate with other hardware that will carry out the application and ensure that the system works [4,5].

The modules, which work with the Arduino microcontroller, measure the EC, pH, DO, and T values of the liquid in the hydroponic system at specified times and transfer them to the microcontroller. The EC sensor can measure between 0 and 20 mS cm^−1^; it supports a broad voltage input of 3–5 V, and its measurement accuracy is ±5%. The pH sensor can measure 0–14 pH; it supports a broad voltage input of 3~5 V, and its measurement accuracy is ±1%. The DO sensor can measure in the range of 0–20 mg L^−1^; it supports a broad voltage input of 3~5 V, and its measurement accuracy is ±2%. Meanwhile, the temperature sensor DS18B20 is a digital temperature sensor; the temperature range it can detect is −55~125 °C, it has an inherent temperature resolution of 0.5 °C, and it supports a broad voltage input of 3~5 V. All sensors, except for the temperature sensors, require maintenance and calibration every six months. There is also a GSM module in the model. This GSM module has Quad-band 850/900/1800/1900 MHz and can connect to any global GSM network with any 2G SIM. Information about the liquid in the system can be obtained at any time through the codes and modules developed by the project coordinator.

In addition to the essential equipment, five submersible pumps and a heater were added to the microcontroller to enable the model to perform its applications. The working principle of the control kit is as follows: 4 different 5 L (replaceable) liquid tanks are connected to the system to perform automation application in line with the data from the EC and pH sensors. In addition, four of the submersible pumps attached to the kit are used in these liquid tanks. In each of the tanks, there is natural spring water to lower the EC value, a fertilizer solution to increase the EC value, a pH-lowering liquid mixture to decrease the pH value, and a pH-increasing liquid mixture to raise the pH value, respectively.

The total production cost of the kit, including all of the hardware and 3D printing costs, is approximately USD 500.

The fertilizer solution was prepared as follows: to prepare 10 L of the fertilizer solution, 30 mL of liquid A and 30 mL of liquid B were used. As a result of the mixing, the nutrients in the 10 L solution in ppm were as follows: nitrate—2060; phosphorus—350; potassium—2700; calcium—1500; magnesium—300; iron—60; manganese—19; boron—7; zinc—2; copper—1; molybdenum—1; the total was approximately 7000 ppm [7,17].

The Arduino microcontroller is operated in a closed box and supplied with voltage from the battery or the outside. It provides electrical energy to the sensors and GSM module as per their needs. Submersible pumps and heaters were fed from a 220 volt mains line. The sensors worked in contact with the water in the system. The kit’s design was carried out with solid modeling software and was made to be closed and compact using a three-dimensional printer. At the same time, using a 3D printer, the kit’s design can be easily integrated into different systems. It can be designed in different sizes and shapes. The processes do not have a specific order and occur continuously in a loop. For example, the heater connected to the kit is used to keep the temperature of the liquid in the system constant in line with the data transmitted by the temperature sensor, which is in contact with the liquid in the system, to the microcontroller. Images of the positioning of the hydroponic system, kit, and sensors installed for the study are given in Figure 1, Figure 2 and Figure 3. Figure 1 shows a visual of the sensors in the kit designed with the help of solid modeling software.

Figure 1 shows the kit designed through solid modeling software and obtained from a 3D printer comprising the box design (1) the submersible pump used to control the dissolved oxygen amount (2), the submersible pump used to decrease the pH value (3), the submersible pump used to increase the pH value (4), the submersible pump used to reduce the EC value (5), another submersible pump (6), the heater (7).

In Figure 2, the final image of the kit, which can be used with the help of solid modeling software and in soilless farming systems, is given—a top view of the control kit that can be used in hydroponic agricultural production systems. The figure comprises the top view of the box design obtained from the 3D printer by using solid modeling software (1), the placement of the dissolved oxygen sensor (2), the location of the electrical conductivity measurement sensor (3), the location of the pH sensor (4), the location of the temperature sensor (5), the battery containment section (6), the submersible pump (7) used to control the dissolved oxygen amount, the submersible pump (8) used to decrease the pH value, the submersible pump (9) used to increase the pH value, the submersible pump (10) used to reduce the EC value, another submersible pump (11), the heater (12), the relay circuit (13), the GSM module (14), and the Arduino nano microcontroller (15) used to increase the EC value.

The features and working principles of the four different submersible pumps used in the four different liquid tanks to change the EC and pH values are as follows: they all have the same features of low flow rate and low power consumption (400 L h^−1^, 4 W). Liquid is not pumped continuously into the system. The desired EC value in the system is 1000 ppm (parts per million) or 1.56 mS cm^−1^. This is associated with the fact that it is the EC value for cultivated plant lettuce [18,19,20,21,22].

The ppm value of the fertilizer solution is approximately 7000 ppm. In this case, every 1 L of fertilizer solution given to the system increases the ppm value of the system by 70 ppm. Therefore, since 100 mL of the liquid can be supplied to the system in 1 s with the liquid pump, the EC value of the system can be increased by 7 ppm or 0.01 ms cm^−1^ in one second.

Natural spring water was used to reduce the EC value of the system. For example, the ppm value of the natural spring water used is 30 ppm (0.04 ms cm^−1^). Each 1 L of natural spring water added to the system reduces the ppm value of the 100 L system by 10 ppm (0.015 ms cm^−1^).

The pH value of the system was set as 6.0, in line with the recommendations of previous studies [18,19,20,21]. Therefore, to reduce the pH of 100 L of water in the existing system by 0.1, 300 mL of pH-reducing liquid must be added. Similarly, to increase the pH of 100 L of water by 0.1, adding 300 mL of pH increaser to the system is sufficient. To change the system’s pH by 0.1, the kit initiates the pump in the tank where the pH regulator solution is located for 3 s and then turns it off.

The amount of dissolved oxygen desired in the system was approximately 6.2 mg L^−1^ [18,19,20,21]. This process was provided by the operation of the air pump, which is in the system and connected to the kit; when the value in the system drops below the desired value and when the expected value is reached, the pump is turned off, and the energy consumption is provided. The air pump used has a capacity of 1000 L h^−1^ and a power of approximately 15 watts.

The liquid in the system, EC, pH, DO, and T values were also monitored with professional test devices, and it was determined that they were in agreement with the values determined by the kit. Therefore, these values are given in the appendices.

The connections and working principle of the developed kit are given in Figure 4. The Arduino nano microcontroller is powered by a 12 V adapter (14), and the ground line is indicated by (15). The sensors connected to the Arduino nano microcontroller use digital and analog pins. The EC sensor (17) is connected to the analog-1 pin (19), the pH sensor (18) is connected to the analog-2 pin (20), the dissolved oxygen sensor (26) is connected to the analog-3 pin (12), and the temperature sensor (16) is connected to the digital-9 pin (6). The GSM module (3) communicates with the Arduino nano microcontroller via the TX (4) and RX (2) pins. Pin-(1) allows the GSM module to be supplied with 5 V-2 A from outside. It is also connected to the ground line with pin-(13). As a result of reading and evaluating the data from the sensors, five different submersible pumps are connected to a relay (34) for the kit to fulfill its function. The relay operates with an external 5 V voltage (36) and is connected to the ground line via pin-(35). The control pins on the relay control each submersible pump separately, turning them on when necessary and turning them off when necessary. There is a pH-reducing submersible pump (21), a pH-increasing submersible pump (22), an EC-reducing submersible pump (23), an EC-increasing submersible pump (24), and one other submersible pump (25) for supplying the required oxygen to the system. Submersible pumps are responsible for delivering the relevant liquids to the system in the desired amount at the desired time. The submersible pumps operate externally and are connected to a 220 V mains line (29, 30, 31, 32, 33). The developed kit also includes a heater (27). The heater is connected to the 220 V mains line (28) and ensures that the liquid in the system remains at the desired temperature. The communication between the relay and Arduino is realized with the digital-10 (5), digital-6 (7), digital-5 (8), digital-4 (9), digital-3 (10), and digital-2 (11) pins.

### Calculation of Power Requirement of the Kit

Since the equipment in the kit will not work continuously, the daily energy consumption and the total cost are calculated as follows [23]:YECET*_k,j_* = ECET*_k,j_* × KM*_k,j_*(1)
(2)$${\mathrm{TYECET}}_{j\;}=\sum_{k=1}^{p}{\mathrm{YECET}}_{k,j}$$

YECET*_k_*_,*j*_ = Annual electrical appliance electricity consumption of a device in the working environment (kWh year^−1^); 

ECET*_k_*_,*j*_ = Electricity consumption for one use of the electrical device in the working environment (kWh usage^−1^); 

KM*_k,j_* = Annual usage of an electrical device in the working environment (usage year^−1^); 

*j*: Working environment; 

TYECET*_k,j_* = Total annual electrical appliance electricity consumption in the working environment (kWh year^−1^); 

*p*: Total number/type of devices in the operating environment.

## 3. Results

A control kit was developed that can be used with all kinds of soilless farming techniques, integrated into any facility of any size, and can control the four different parameters of the liquid in the system, which are considered the most important for soilless agriculture. This kit could control the water parameters in the system with its inexpensive Arduino microcontroller; hardware; and original, open-source software. After the data acquisition phase, the kit can intervene in the system and ensure that the parameters reach the desired levels.

According to the results obtained and the measurements made, the electricity consumption of the model is a maximum of 35 watts h^−1^, a minimum of 3 watts h^−1^, and an average energy consumption of 4 watts h^−1^. Therefore, the daily electrical energy consumption is approximately 0.096 kWh. In this case, the daily electrical energy cost of the model is USD 0.0137 and the monthly electrical energy cost is USD 0.411.

The graphs regarding the reading values obtained for various parameters of the liquid in the system are given below.

During the study, the values obtained from the liquid in the system with the help of professional devices were compared with those obtained simultaneously from the developed kit. For this, the intraclass correlation coefficient (ICC) was found using a one-way random, single-measures method. ICC can be used when quantitative measurements are made on units organized into groups, and it describes how strongly units in the same group resemble each other.

According to the statistical analysis from the data obtained, a robust linear relationship was found between EC and pH (R^2^ = 0.99). Similarly, a strong linear relationship was obtained between EC and DO values (R^2^ = 0.99) shown Figure 5. Additionally, linear relationships were found between the EC values in the system and total N, with R^2^ = 0.80; EC and Fe, R^2^ = 0.48 shown Figure 6; EC and K, R^2^ = 0.84; EC and Mg, R^2^ = 0.86 shown Figure 7; and EC and PO_4_, R^2^ = 0.94 and Figure 8.

Linear relationships were found between the values obtained from the professional tester and the values obtained from the kit for the EC value with R^2^ = 0.95, between the values obtained from the professional tester and the values obtained from the kit for the pH value with R^2^ = 0.90, and between the values obtained from the professional tester and the values obtained from the kit for DO values with R^2^ = 0.91. According to these results, it is understood that the values obtained from the model can be used. The values obtained by the automation kits from the liquid in the hydroponic system during the growing period of a lettuce plant are given in Table 1.

To test the system, two different lettuce types called “Lollo rosso green leaf” were grown for 30 days under the same conditions. Approximately 200 lettuces were grown in a 12 m^2^ area; 100 were grown without the developed kit, and 100 were grown using the developed kit. The conditions in the environment were the same: 22 °C, 39% humidity, 165.6 µmol s^−1^ artificial lighting intensity, and 12 h day^−1^ of artificial lighting. The average yield of lettuce plants grown with the developed kit was 2.11 kg m^2^, while the average yield of lettuce plants grown using only human labor was 1.75 kg m^2^. This means that an approximately 21% yield increase can be achieved. It also means a reduction in the labor force.

## 4. Discussion

Developments in hydroponic systems continue and are implemented worldwide. The system in this paper was developed for all kinds of soilless farming techniques; to be integrated into facilities of any size; and to control four different parameters of the liquid in the system, which are considered essential for agriculture with soilless farming techniques. The control kit can control the water parameters in the system with its inexpensive Arduino microcontroller, hardware, and software developed in the original open-source C++ software language; thus, it is open to development. Since the control kit is open-source, it can be modified by the user and integrated into any size of system. The design can provide the parameters to reach the desired level by interfering with the liquid in the soilless agriculture system. Additionally, data can be monitored from smartphones.

Hydroponics increases production costs due to high capital investments and high energy requirements. These high costs are acceptable only when grown crops have higher quality characteristics. However, the high production costs of hydroponics make it unfeasible to economically grow staple crops (maize, soybeans, wheat, rice, potatoes, etc.). Therefore, hydroponics is more suited to producing green crops and some fruit-bearing crops (e.g., strawberries, tomatoes, and peppers) [24,25].

Labor is the costliest factor for the grower in hydroponic farming [26]. Hydroponics is significantly more labor-intensive than traditional field farming and lacks widespread automation practices. Without significant advances in automation, human labor will remain important. Hydroponic farms operate year-round; so, local labor is preferable to traditional farms that employ seasonal workers. Since most of these enterprises are located in urban areas, they usually have access to local labor. However, workers from urban areas typically need more experience or training for the labor required for the enterprises [27].

Automation practices in hydroponics can reduce some of the labor costs. Automated systems can control the planting, transplanting, transportation, irrigation, harvesting, and post-harvest operations. Data collection and monitoring of environmental parameters enable the optimization of soilless farming systems. The most important advantage of these systems is data-driven decision-making and implementation. These data include air temperature, root zone temperature, humidity/vapor pressure deficit, CO_2_ concentration, dissolved O_2_, light intensity, nutrient concentration, pH, etc. [28].

In a study conducted in 2016, an embedded system that can be applied to livestock was developed. The system provided a solution to track cattle feeding activities and their separation from rumination activity. The answer is inspired by the solutions designed for humans. In the proposed setup, a popular piece of data acquisition hardware that contained two sensors was used in human-related applications. This hardware was a microcontroller board with two acceleration sensors, one directional and one angular. The card detected accelerations in three planes and directional movements. It can detect angular changes by detecting angular accelerations in three planes [29]. The cited study for livestock showed that inexpensive microcontrollers such as Arduino can be used in agriculture.

Another study conducted in 2019 focused on meteorological data tracking systems, and a tracking system was developed using the “Internet of Things” (IoT) method [30]. Meteorological data related to the meteorological data tracking station installed on the roof of Kocaeli University were made available to all users with Internet access by being transferred to the cloud environment. The installed system monitored global radiation, wind speed, wind direction, temperature, and humidity data. According to the data obtained from the system, the monthly total radiation amounts per square meter were found to be 195 kWh m^−2^, 174 kWh m^−2^, 134 kWh m^−2^, 86 kWh m^−2^, and 79 kWh m^−2^ for August, September, October, November, and December, respectively [30]. As a result of this study, it was observed that the data could be tracked using the Internet. Furthermore, the control kit obtained from the study can perform data transfer using the GSM network.

Rani and Kamalesh developed an automatic irrigation system using an Arduino microcontroller [31]. In the research, water flow was allowed if a specific humidity level in the work area was reached. The water pressure was updated by considering the irrigation pipes and flow rate. Information on soil moisture and irrigation could be accessed via the system’s website. The farmer could check the water level and engine status whenever they wanted, and an informational message was sent to their mobile phone [31]. Additionally, by evaluating the data obtained from different sensors in the control kit obtained as a result of the study, it could control the parameters of the liquid in the soilless agriculture system.

Dhatri et al. developed a soil moisture sensor that transmits data according to the state of the soil and works with Arduino [32]. One of the primary purposes of the study was to use soil moisture sensors and low-cost Arduino-based automatic irrigation systems to help the agricultural irrigation process.

The agricultural sector is one of the most critical sectors in Malaysia and the world, and providing an optimal environment for raising high-quality poultry is necessary. Chicken farms are one of the vital sectors of the food supply chain. For example, the “Internet of Things Application” can be employed to develop active surveillance systems that demand optimum values such as ammonia, carbon dioxide, humidity, and temperature. One study researched the development of a “smart poultry farm.” Ammonia and carbon dioxide levels were determined using various sensors and adjusted to optimum levels using a discharge fan. Humidity and temperature values were also similarly measured and controlled [33]. As a result of experiments, it was shown that the proposed system had better monitoring and control than traditional Arduino systems. As a result of the study, it was determined that the Arduino microcontroller and hardware could be used efficiently for remote monitoring, remote data recording, and remote control.

Nayyar and Vikram developed a system to enable farmers to practice smart agriculture and obtain live data (temperature and soil moisture) to improve their products’ overall yield and quality [34]. The IoT-based software they developed was integrated with Arduino technology and was supported by various sensors. The data obtained could be monitored instantly online. The proposed system was tested and worked with over 98% accuracy.

A low-cost, energy-efficient drip irrigation system was developed [35]. This design can be used in large agricultural areas and small gardens. The developed system is an intelligent drip irrigation system with ultrasonic sensors and solenoid valves. The system is fully automatic, and communication can be conducted via e-mail. However, it has been reported that the system should be tested manually to determine any failure in the hardware.

Vincentdo et al. developed an automatic hydroponic monitoring and controlling system for conventional and hydroponic farmers. As a result of this study, it was reported that the system could provide better growth of plants by accurately adjusting pH and nutrient levels [16].

A 2018 study proposed developing an intelligent IoT-based hydroponic system. The developed system is reported to be able to control pH, temperature, humidity, and lighting intensity parameters. These parameters were collected in real time for weeks and were reported to be obtained with an accuracy of 88% [36].

In another study conducted in 2019, a system was developed for greenhouses to monitor and control parameters such as light intensity, pH, electrical conductivity, water temperature, and relative humidity. According to the results obtained, it was reported that the system works with 84.53% accuracy [37].

A 2016 study aimed to determine the environment’s temperature, humidity, and light intensity for agricultural activities carried out in closed environments and to control them appropriately. The control network tested in tomato cultivation operated in the 400 MHz band, and IEEE 802.15.6 Wi-Fi standard was reported to work reliably [38].

In this study, IoT-based smart greenhouse automation was developed using Arduino. This study aimed to increase the efficiency of the greenhouse. The Arduino microcontroller and appropriate software and sensors were used in the system. The microcontroller used was Arduino Mega, and the sensors were intended to measure the greenhouse’s soil moisture and light intensity for irrigation purposes. If the temperature rises above the set limit, a fan is automatically started as a cooler to lower the temperature [39].

The kit developed in the present study was an automatic control system employing a microcontroller, various sensors, and appropriate open-source and original software. It can keep its electrical conductivity, acidity or alkalinity, dissolved oxygen, and temperature values within the desired range. It can provide information about the system to the user. Developments in hydroponic systems continue and are implemented worldwide. The system presented in this paper was developed to be adapted to all kinds of soilless farming techniques; to be integrated into facilities of any size; and to control four different parameters of the liquid in the system, which are considered the most important for agriculture with soilless farming techniques. The invention can control the water parameters in the system with its inexpensive Arduino microcontroller, hardware, and software developed in the original open-source C++ software language; thus, it is open to development. C++ is a widely used, general-purpose programming language. In 1983, the name was changed to C++. Since the invention is open-source, it can be modified by the user and integrated into systems of any size. The developed control kit can provide the parameters to reach the desired level by interfering with the liquid in the soilless agriculture system. Additionally, data can be monitored from smartphones.

## 5. Conclusions

The study yielded the following conclusions:

Automation can be utilized in soilless agricultural production systems, which are rapidly becoming widespread and gaining importance worldwide, where even land that cannot be used for agriculture can be used. All parameters available in agricultural production can be controlled. Technology, hardware, and software are used intensively in such a system.

The control kit developed in the present study determined that plants could be grown successfully using soilless or vertical farming techniques.

It was determined that Arduino or similar affordable and easy-to-use open-source microcontrollers could be used in agriculture.

## 6. Patents

The author owns the pending utility model/patent application in Turkey. The application number is 2022/017335, and the document number is 2022-GE-830423.

## Figures and Tables

**Figure 1 sensors-23-03706-f001:**
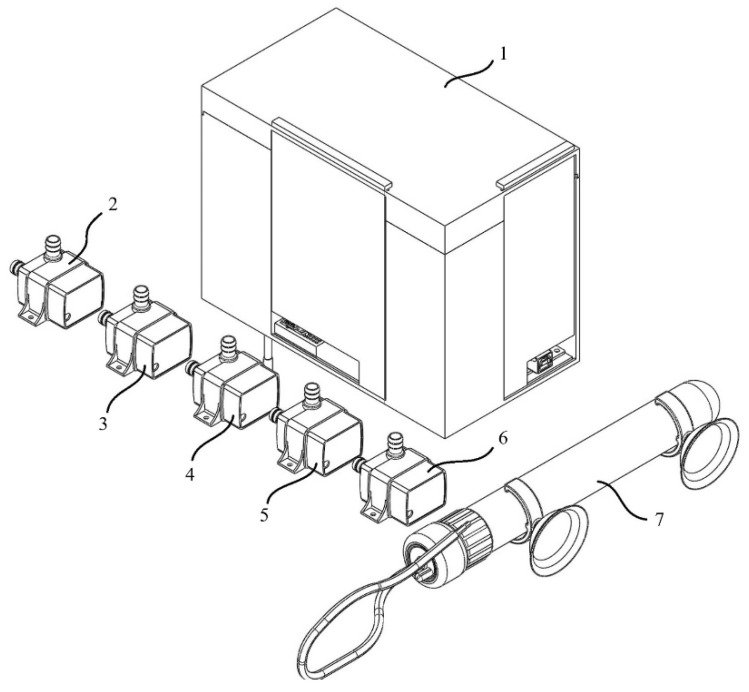
Images of the kit designed with the help of solid modeling software. Box design (1) the submersible pump used to control the dissolved oxygen amount (2), the submersible pump used to decrease the pH value (3), the submersible pump used to increase the pH value (4), the submersible pump used to reduce the EC value (5), another submersible pump (6), the heater (7).

**Figure 2 sensors-23-03706-f002:**
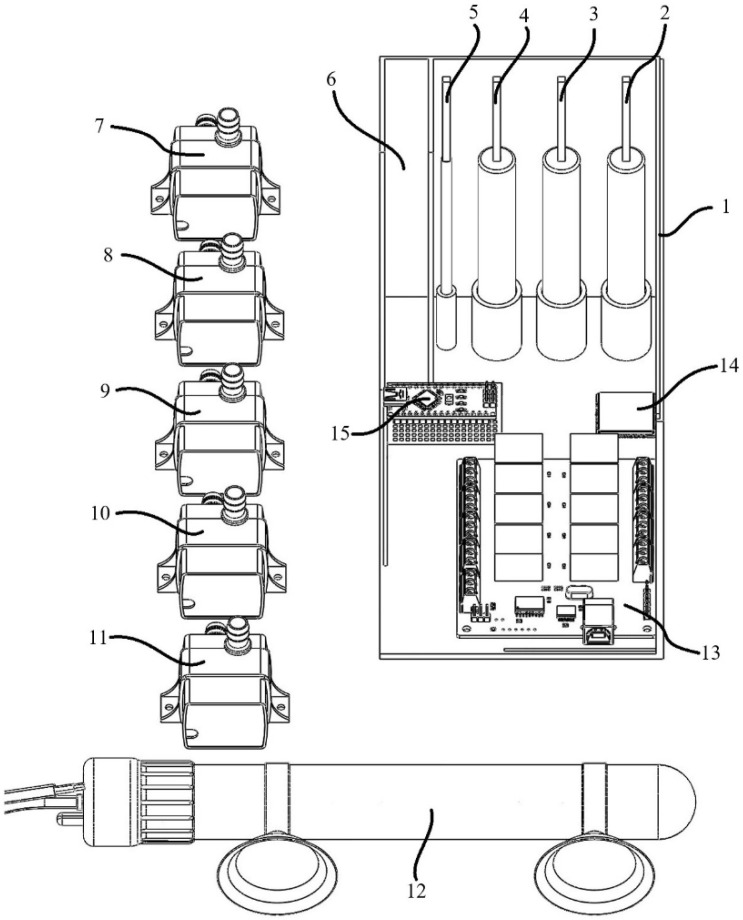
Images of the kit. Box design (1), the placement of the dissolved oxygen sensor (2), the location of the electrical conductivity measurement sensor (3), the location of the pH sensor (4), the location of the temperature sensor (5), the battery housing section (6), the submersible pump used to control the dissolved oxygen amount (7), the submersible pump used to decrease the pH value (8), the submersible pump used to increase the pH value (9), the submersible pump used to reduce the EC value (10), another submersible pump (11), the heater (12), the relay circuit (13), the GSM module (14), and the Arduino nano microcontroller (15).

**Figure 3 sensors-23-03706-f003:**
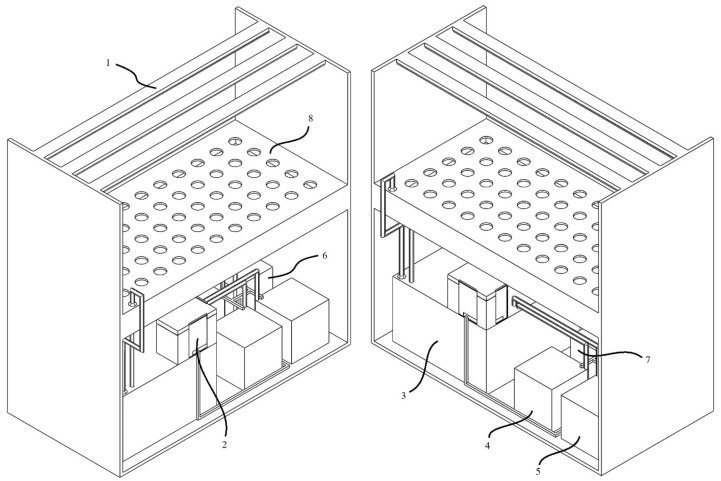
Positioning of the hydroponic system and the kit. (1) Artificial lighting unit; (2) the kit obtained as a result of the project; (3) liquid reservoir; (4) EC-increasing fertilizer solution; (5) pH-increasing solution; (6) EC-lowering solution; (7) pH-lowering solution; (8) cultivation environment.

**Figure 4 sensors-23-03706-f004:**
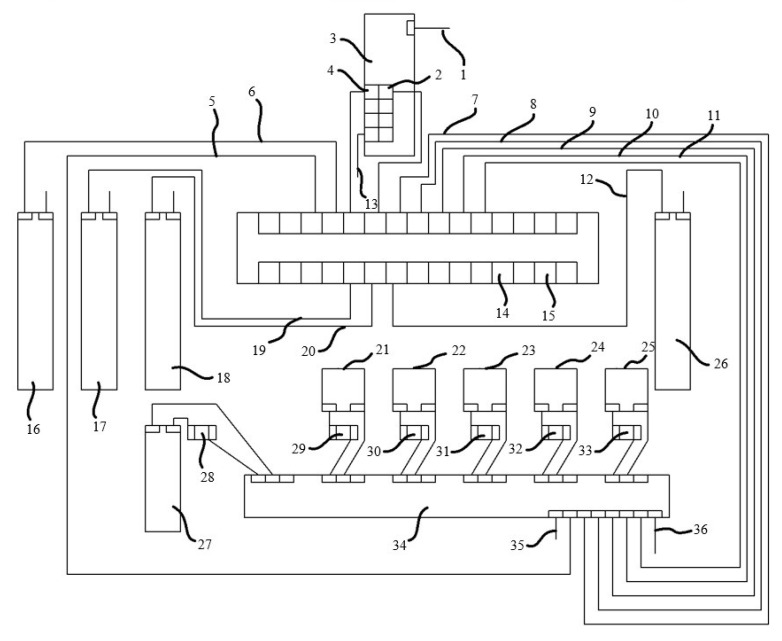
Demonstration of the connection and operation of the developed automation kit.

**Figure 5 sensors-23-03706-f005:**
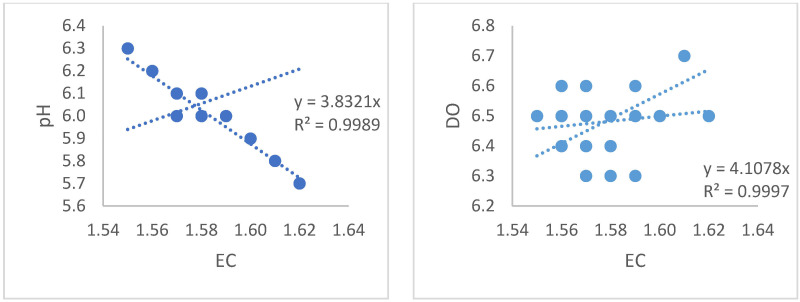
Relationships between the EC values and the pH and DO values of the liquid in the system.

**Figure 6 sensors-23-03706-f006:**
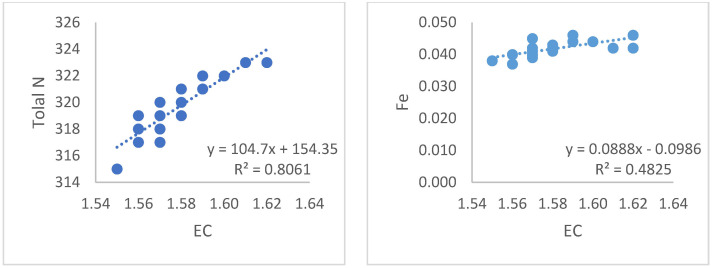
Relationships between the EC values of the liquid in the system and the total N and Fe values.

**Figure 7 sensors-23-03706-f007:**
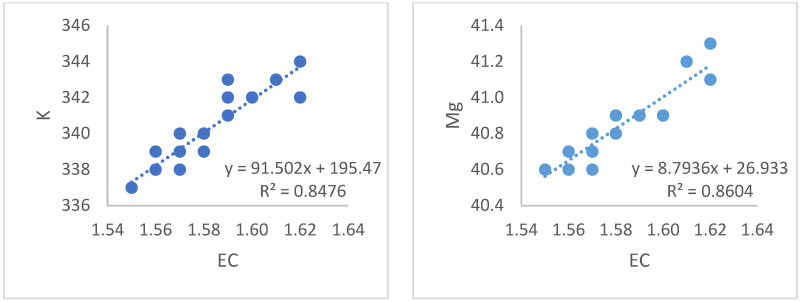
Relationships between the EC values and the K and Mg values of the liquid in the system.

**Figure 8 sensors-23-03706-f008:**
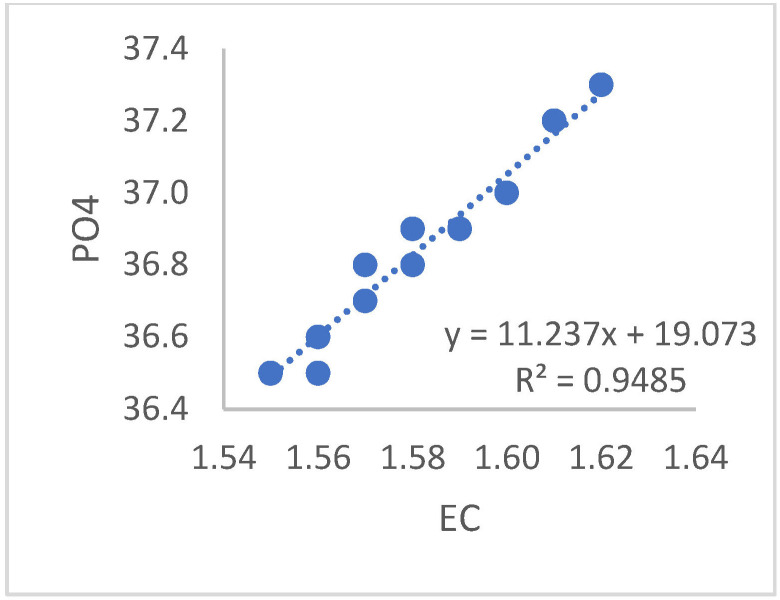
Relationship between the EC values and PO_4_ values of the liquid in the system.

**Table 1 sensors-23-03706-t001:** Specific parameters of the liquid in the system.

Days	EC (mS cm^−1^) (Values Obtained from the Professional Test Kit)	EC (mS cm^−1^) (Values Obtained from the Developed Kit)	pH (Values Obtained from the Professional Test Kit)	pH (Values Obtained from the Developed Kit)	DO (mg L^−1^) (Values Obtained from the Professional Test Kit)	DO (mg L^−1^) (Values Obtained from the Developed Kit)
1	1.59	1.59	6.0	6.0	6.5	6.5
2	1.57	1.57	6.1	6.1	6.4	6.4
3	1.59	1.58	6.0	6.0	6.3	6.3
4	1.56	1.57	6.2	6.2	6.4	6.4
5	1.56	1.56	6.2	6.2	6.6	6.6
6	1.56	1.56	6.2	6.1	6.5	6.5
7	1.57	1.57	6.1	6.1	6.5	6.5
8	1.59	1.59	6.0	6.0	6.5	6.5
9	1.57	1.57	6.1	6.1	6.4	6.4
10	1.56	1.56	6.2	6.2	6.6	6.6
11	1.57	1.56	6.1	6.0	6.4	6.4
12	1.57	1.57	6.1	6.1	6.3	6.3
13	1.58	1.57	6.0	6.1	6.3	6.3
14	1.56	1.56	6.2	6.2	6.4	6.3
15	1.62	1.62	5.7	5.7	6.5	6.5
16	1.57	1.57	6.1	6.1	6.6	6.6
17	1.58	1.58	6.1	6.1	6.4	6.4
18	1.55	1.55	6.3	6.2	6.5	6.5
19	1.57	1.57	6.1	6.1	6.4	6.5
20	1.59	1.59	6.0	6.0	6.5	6.5
21	1.57	1.57	6.1	6.1	6.4	6.4
22	1.59	1.58	6.0	6.0	6.6	6.6
23	1.57	1.57	6.0	6.1	6.6	6.6
24	1.57	1.57	6.0	6.0	6.5	6.5
25	1.60	1.60	5.9	5.9	6.5	6.4
26	1.57	1.57	6.1	6.1	6.5	6.5
27	1.58	1.58	6.0	6.0	6.5	6.5
28	1.57	1.57	6.0	6.0	6.6	6.6
29	1.61	1.61	5.8	5.8	6.7	6.7
30	1.62	1.62	5.7	5.8	6.5	6.5

## Data Availability

Data is unavailable due to privacy.

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
