# Peer review of "Development of an Embedded Software and Control Kit to Be Used in Soilless Agriculture Production Systems"

_sensors, 2023, doi:10.3390/s23073706_

Round 1
Reviewer 1 Report
Review comments for paper Sensors-2264065
Development of an Embedded Software and Control Kit to be used in Soilless Agriculture Production Systems.
This manuscript is about the development of an Embedded Software and Control Kit to be used in Soilless Agriculture Production Systems. This manuscript needs careful modification. I feel that the work submitted is an important work in the field of water saving and soilless agriculture production systems. This work can be recommended for publication after minor revision, improvement and simplification.
Some comments could be made and these are summarized below:
1. Considering amount of work with similar topics have been published in several journals. Please explain the originality and novelty of this research work, compared with similar works?
2. Please, the objectives of this work should be clearer.
3. The discussion of the obtained results must improved.
4. Finally, the introduction must be improved.

Author Response
Response to Reviewer#1:
Suggested revisions by Reviewer#1
|
1. Considering amount of work with similar topics have been published in several journals. Please explain the originality and novelty of this research work, compared with similar works? |
|
2. Please, the objectives of this work should be clearer. |
|
3. The discussion of the obtained results must improved. |
|
4. Finally, the introduction must be improved. |
Corrections given by Reviewer#1
This article has undergone English language editing by MDPI. The text has been checked for correct use of grammar and common technical terms, and edited to a level suitable for reporting research in a scholarly journal.
|
1. line 93 to 114 Today, automation applications in agriculture and soilless agriculture have become widespread. The present study aimed to fill in some of the gaps encountered in these applications. While automation applications control specific parameters, IoT applications aim to determine environmental factors and exchange data. In modern agriculture, IoT technologies are often used to collect real-time information such as the temperature, humidity, wind, weather, rainfall, soil moisture, soil electrical conductivity, pH value, and soil nitrogen value. Smart agriculture is defined as the interactive operation of many technologies for agricultural purposes [19]. With IoT, the workload in innovative agriculture technologies can be reduced while at the same time increasing the yield and quality of the product. Many operations can be realized in IoT-based agricultural applications. Guo and Zhong applied IoT techniques to precise irrigation, fertilization, and air conditioning systems for more efficient greenhouse production [14]. Srbinaovska et al. proposed a wireless sensor network architecture for vegetable production in greenhouses and reduced management costs by monitoring environmental data [32]. These researchers designed a practical and cost-effective data monitoring system based on a wireless sensor network technology to monitor basic environmental parameters such as temperature, humidity, and light. Türker et al. developed a prototype system to measure, wirelessly transmit, and monitor the temperature changes inside a greenhouse using the IoT method. The system consists of Raspberry Pi, a Wi-Fi adapter, and two DS18B20 temperature sensors. The system records air, soil, and water temperature data in a greenhouse and transmits them to a database on the web [34]. Vincentdo and Surantha studied the control of pH and plant nutrients in soilless agriculture [36]. |
|
2. line 114 to 119 The present study aimed to provide the instant control of four different parameters and transfer data to the user. Examining the previous studies, it was seen that there are studies in which only pH control or only plant nutrient control or only pH and plant nutrient control or only temperature and pH control were used, whereas short message services were not. No device can instantly control all of these parameters and transmit information to the user. |
|
3. Obtained results improved, Table 2 added. |
|
4. Introduction improved. Line 93 to 119 added
Today, automation applications in agriculture and soilless agriculture have be-come widespread. The present study aimed to fill in some of the gaps encountered in these applications. While automation applications control specific parameters, IoT ap-plications aim to determine environmental factors and exchange data. In modern ag-riculture, IoT technologies are often used to collect real-time information such as the temperature, humidity, wind, weather, rainfall, soil moisture, soil electrical conduc-tivity, pH value, and soil nitrogen value. Smart agriculture is defined as the interactive operation of many technologies for agricultural purposes [19]. With IoT, the workload in innovative agriculture technologies can be reduced while at the same time increasing the yield and quality of the product. Many operations can be realized in IoT-based agricultural applications. Guo and Zhong applied IoT techniques to precise irrigation, fertilization, and air conditioning systems for more efficient greenhouse production [14]. Srbinaovska et al. proposed a wireless sensor network architecture for vegetable production in greenhouses and reduced management costs by monitoring environ-mental data [32]. These researchers designed a practical and cost-effective data moni-toring system based on a wireless sensor network technology to monitor basic envi-ronmental parameters such as temperature, humidity, and light. Türker et al. devel-oped a prototype system to measure, wirelessly transmit, and monitor the temperature changes inside a greenhouse using the IoT method. The system consists of Raspberry Pi, a Wi-Fi adapter, and two DS18B20 temperature sensors. The system records air, soil, and water temperature data in a greenhouse and transmits them to a database on the web [34]. Vincentdo and Surantha studied the control of pH and plant nutrients in soilless agriculture [36]. The present study aimed to provide the instant con-trol of four different parameters and transfer data to the user. Examining the previous studies, it was seen that there are studies in which only pH control or only plant nu-trient control or only pH and plant nutrient control or only temperature and pH con-trol were used, whereas short message services were not. No device can instantly con-trol all of these parameters and transmit information to the user. |

Reviewer 2 Report
1. Add take home message at the end of the abstract or clarify the experimental site and conditions in the abstract 2. Introduction section is well written but lacks relevant data. Add more relevant review of literature in the introduction section also elaborate the benefits of soilless culture -Research gaps are not well identified. Identify research gaps by using recent data. -Clarify the hypothesis at the end of introduction section. 3. References needs to be cross matched 4. It is better to draw schematic diagram and avoid complex figures making them simple and easy to understand with elaboration 5. Discussion part is quite lengthy make it more relevant with accurate information 6. Reference should be updated for last six years 7. References Format as per journal guidelines Overall Language English language needs a through revision from some professional service or native speaker. I will suggest the author to work on above mentioned shortcomings and after revision manuscript can be accepted for publication.
Author Response
Response to Reviewer#2:
Suggested revisions by Reviewer#2
|
1. Add take home message at the end of the abstract or clarify the experimental site and conditions in the abstract |
|
2. Introduction section is well written but lacks relevant data. Add more relevant review of literature in the introduction section also elaborate the benefits of soilless culture -Research gaps are not well identified. Identify research gaps by using recent data. -Clarify the hypothesis at the end of introduction section. |
|
3. References needs to be cross matched |
|
4. It is better to draw schematic diagram and avoid complex figures making them simple and easy to understand with elaboration |
|
5. Discussion part is quite lengthy make it more relevant with accurate information |
|
6. Reference should be updated for last six years |
|
7. References Format as per journal guidelines Overall Language English language needs a through revision from some professional service or native speaker. I will suggest the author to work on above mentioned shortcomings and after revision manuscript can be accepted for publication. |
Corrections given by Reviewer#2
This article has undergone English language editing by MDPI. The text has been checked for correct use of grammar and common technical terms, and edited to a level suitable for reporting research in a scholarly journal.
|
1. Added in Abstract section (line 24 to 28).
The area where the experiment was conducted is a fully controlled and closed area established within Ankara University. The ambient temperature is 22°C, and the humidity is 39%. The coordinates of the experimental area are 39.962013 and 32.867491. Three different artificial lighting intensities (165.6 µmol m-2 s-1, 248.4 µmol m-2 s-1, and 331.2 µmol m-2 s-1) and a desired photoperiod duration can be applied to the site. |
|
2. I clarify the hypothesis at the end of introduction section and added to more review of literature (line 93 to 119)
Today, automation applications in agriculture and soilless agriculture have become widespread. The present study aimed to fill in some of the gaps encountered in these applications. While automation applications control specific parameters, IoT applications aim to determine environmental factors and exchange data. In modern agriculture, IoT technologies are often used to collect real-time information such as the temperature, humidity, wind, weather, rainfall, soil moisture, soil electrical conductivity, pH value, and soil nitrogen value. Smart agriculture is defined as the interactive operation of many technologies for agricultural purposes [19]. With IoT, the workload in innovative agriculture technologies can be reduced while at the same time increasing the yield and quality of the product. Many operations can be realized in IoT-based agricultural applications. Guo and Zhong applied IoT techniques to precise irrigation, fertilization, and air conditioning systems for more efficient greenhouse production [14]. Srbinaovska et al. proposed a wireless sensor network architecture for vegetable production in greenhouses and reduced management costs by monitoring environmental data [32]. These researchers designed a practical and cost-effective data monitoring system based on a wireless sensor network technology to monitor basic environmental parameters such as temperature, humidity, and light. Türker et al. developed a prototype system to measure, wirelessly transmit, and monitor the temperature changes inside a greenhouse using the IoT method. The system consists of Raspberry Pi, a Wi-Fi adapter, and two DS18B20 temperature sensors. The system records air, soil, and water temperature data in a greenhouse and transmits them to a database on the web [34]. Vincentdo and Surantha studied the control of pH and plant nutrients in soilless agriculture [36]. The present study aimed to provide the instant control of four different parameters and transfer data to the user. Examining the previous studies, it was seen that there are studies in which only pH control or only plant nutrient control or only pH and plant nutrient control or only temperature and pH control were used, whereas short message services were not. No device can instantly control all of these parameters and transmit information to the user. |
|
3. All the references checked and new references added. |
|
4. I draw a schematic diagram, added and explained (Figure 4) |
|
5. Discussion part improved (line 439 to 460).
Vincentdo et al. developed an automatic hydroponic monitoring and controlling system for conventional and hydroponic farmers. As a result of this study, it was re-ported that the system could provide better growth of plants by accurately adjusting pH and nutrient levels [36]. A 2018 study proposed developing an intelligent IoT-based hydroponic system. The developed system is reported to be able to control pH, temperature, humidity, and lighting intensity parameters. These parameters were collected in real time for weeks and were reported to be obtained with an accuracy of 88% [23]. In another study conducted in 2019, a system was developed for greenhouses to monitor and control parameters such as light intensity, pH, electrical conductivity, water temperature, and relative humidity. According to the results obtained, it was reported that the system works with 84.53% accuracy [2]. A 2016 study aimed to determine the environment's temperature, humidity, and light intensity for agricultural activities carried out in closed environments and to control them appropriately. The control network tested in tomato cultivation was 400 MHz band, and IEEE 802.15.6 Wi-Fi standard was reported to work reliably [18]. In this study, IoT-based smart greenhouse automation was developed using Ar-duino. This study aimed to increase the efficiency of the greenhouse. Arduino micro-controller and appropriate software and sensors were used in the system. The micro-controller used was Arduino Mega, and the sensors were intended to measure the greenhouse's soil moisture and light intensity for irrigation purposes. If the tempera-ture rises above the set limit, a fan is automatically started as a cooler to lower the temperature [31]. |
|
6. New references added (2-8-14-18-19-23-31-32-34-36-39) |
|
7. English language revised.
This article has undergone English language editing by MDPI. The text has been checked for correct use of grammar and common technical terms, and edited to a level suitable for reporting research in a scholarly journal. |

Reviewer 3 Report
Dear Author,
The manuscript entitled: Development of an Embedded Software and Control Kit to be Used in Soilless Agriculture Production Systems, aimed to develop an automatic control system and tested employing a microcontroller, various sensors, appropriate open-source codes, and original software. Electrical conductivity (EC), Power of Hydrogen (pH), dissolved oxygen (DO), and temperature (T) values were determined by the developed control system.
The manuscript presents a disruptive proposal, with the possibility of application in soilless farming, low cost, and showed some results of the system tests. However, the manuscript needs a thorough review by the author, as important information and clarifications are missing in the methodology and results of the study. I have made my comments and recommendations and text box along in the attached file. The author must answer and review them, and I highlight the main ones as follows:
a) The covariates used in the tests (EC, pH, OD and T) must be described, justifying their importance, and the correlations with the EC must also be described and justified;
b) The author must make clear the objectives of the study;
c) The Material and Methods topic lacks a lot of information that the author should insert, including methodologies that appear in the Results, but that were not described;
d) Considering that the author has submitted this innovation for patent registration, the author will be able to detail the information with the specifications of the components used (hardware), as well as the line of programming (C++) used in the study;
e) I recommend that the author include the cost of each kit component, including 3D printing;
f) The author must make it clear if the system prototype was built, how and where it was used, the experimental design used (measurement repetitions and time required).
The manuscript is relevant for use in soiless agriculture activities, in the sense of technological innovation, increases in productivity and profits, with cost reduction. I recommend a major revision of the manuscript, with the possibility of publication after these adjustments are reviewed by the author.

Author Response
Response to Reviewer#3:
Suggested revisions by Reviewer#3
|
a) The covariates used in the tests (EC, pH, OD and T) must be described, justifying their importance, and the correlations with the EC must also be described and justified; |
|
b) The author must make clear the objectives of the study; |
|
c) The Material and Methods topic lacks a lot of information that the author should insert, including methodologies that appear in the Results, but that were not described; |
|
d) Considering that the author has submitted this innovation for patent registration, the author will be able to detail the information with the specifications of the components used (hardware), as well as the line of programming (C++) used in the study; |
|
e) I recommend that the author include the cost of each kit component, including 3D printing; |
|
f) The author must make it clear if the system prototype was built, how and where it was used, the experimental design used (measurement repetitions and time required). |
Corrections given by Reviewer#3
|
a) The covariates described (line 173 to 177, line 222 to 241)
The fertilizer solution was prepared as follows: to prepare 10 L of the fertilizer solution, 30 mL of liquid A and 30 mL of Liquid B were used. As a result of the mixing, the nutrients in the 10 L solution in ppm were as follows: nitrate: 2060; phosphorus: 350; potassium: 2700; calcium: 1500; magnesium: 300; iron: 60; manganese: 19; boron: 7; zinc: 2; copper: 1; molybdenum: 1, and the total was approximately 7000 ppm [11,28].
The desired EC value in the system is 1000 ppm (parts per million) or 1.56 ms cm-1. This was associated with the fact that this is the EC value for the cultivated plant lettuce. [10,15,17,27]. The ppm value of the fertilizer solution is approximately 7000 ppm. In this case, every 1 L of fertilizer solution given to the system increases the ppm value of the system by 70 ppm. Therefore, since 100 mL of the liquid can be supplied to the system in 1 second with the liquid pump, the EC value of the system can be increased by 7 ppm or 0.01 ms cm-1 in one second. Natural spring water was used to reduce the EC value of the system. For example, the ppm value of the natural spring water used is 30 ppm (0.04 ms cm-1). Each 1 L of natural spring water added to the system reduces the ppm value of the 100 L system by 10 ppm (0.015 ms cm-1). The pH value of the system was set as 6.0, in line with the recommendations of previous studies [10,15,17,27]. Therefore, to reduce the pH of 100 L of water in the existing system by 0.1, 300 mL of pH-reducing liquid must be added. Similarly, to increase the pH of 100 L of water by 0.1, adding 300 mL of pH increaser to the system is sufficient. To change the system's pH by 0.1, the kit initiates the pump in the tank where the pH regulator solution is located for 3 seconds and then turns it off.
|
|
b) Objectives of the study expanded at and of the introduction section (line 93 to 129)
Today, automation applications in agriculture and soilless agriculture have become widespread. The present study aimed to fill in some of the gaps encountered in these applications. While automation applications control specific parameters, IoT applications aim to determine environmental factors and exchange data. In modern agriculture, IoT technologies are often used to collect real-time information such as the temperature, humidity, wind, weather, rainfall, soil moisture, soil electrical conductivity, pH value, and soil nitrogen value. Smart agriculture is defined as the interactive operation of many technologies for agricultural purposes [19]. With IoT, the workload in innovative agriculture technologies can be reduced while at the same time increasing the yield and quality of the product. Many operations can be realized in IoT-based agricultural applications. Guo and Zhong applied IoT techniques to precise irrigation, fertilization, and air conditioning systems for more efficient greenhouse production [14]. Srbinaovska et al. proposed a wireless sensor network architecture for vegetable production in greenhouses and reduced management costs by monitoring environmental data [32]. These researchers designed a practical and cost-effective data monitoring system based on a wireless sensor network technology to monitor basic environmental parameters such as temperature, humidity, and light. Türker et al. developed a prototype system to measure, wirelessly transmit, and monitor the temperature changes inside a greenhouse using the IoT method. The system consists of Raspberry Pi, a Wi-Fi adapter, and two DS18B20 temperature sensors. The system records air, soil, and water temperature data in a greenhouse and transmits them to a database on the web [34]. Vincentdo and Surantha studied the control of pH and plant nutrients in soilless agriculture [36]. The present study aimed to provide the instant control of four different parameters and transfer data to the user. Examining the previous studies, it was seen that there are studies in which only pH control or only plant nutrient control or only pH and plant nutrient control or only temperature and pH control were used, whereas short message services were not. No device can instantly control all of these parameters and transmit information to the user. Automation kits used in soilless farming systems are high-cost test devices. Therefore, the control kit obtained within the project's scope will be fast and easy to use, allowing it to become widespread due to its affordable price and easy use. Therefore, a control kit was developed with an Arduino microcontroller, four different sensors, auxiliary equipment, and software prepared by the project coordinator. Although such controller cards are used in many sectors today, their use in agriculture is not at the desired level. Furthermore, today, no device can simultaneously control the electrical conductivity, pH, dissolved oxygen, and temperature of the solution in systems cultivated with the soilless farming technique. The present study was conducted to find a solution to the needs in this area.
|
|
c) Methodologies inserted at the Material and Methods (line 253 to 279 and added figure 4)
The connections and working principle of the developed kit are given in Figure 4. The Arduino nano microcontroller is powered by a 12 V adapter (14), and the ground line is indicated by 15. The sensors connected to the Arduino nano microcontroller use digital and analog pins. The EC sensor (17) is connected to the analog-1 pin (19), the pH sensor (18) is connected to the analog-2 pin (20), the dissolved oxygen sensor (26) is connected to the analog-3 pin (12), and the temperature sensor (16) is connected to the digital 9-pin (6). The GSM module (3) communicates with the Arduino nano microcontroller via the TX (4) and RX (2) pins. Pin-1 allows the GSM module to be supplied with 5 V-2 A from outside. It is also connected to the ground line with pin-13. As a result of reading and evaluating the data from the sensors, five different submersible pumps are connected to a relay (34) for the kit to fulfill its function. The relay operates with an external 5 V voltage (36) and is connected to the ground line via pin-35. The control pins on the relay control each submersible pump separately, turning them on when necessary and turning them off when necessary. There is a pH-reducing submersible pump (21), a pH-increasing submersible pump (22), an EC-reducing submersible pump (23), an EC-increasing submersible pump (24), and one other submersible pump (25) for supplying the required oxygen to the system. Submersible pumps are responsible for delivering the relevant liquids to the system in the desired amount at the desired time. The submersible pumps operate externally and are connected to a 220 V mains line (29, 30, 31, 32, 33). The developed kit also includes a heater (27). The heater is connected to the 220 V mains line (28) and ensures that the liquid in the system remains at the desired temperature. The communication between the relay and Arduino is realized with the digital-10 (5), digital-6 (7), digital-5 (8), digital-4 (9), digital-3 (10), and digital-2 (11) pins. |
|
d) Detail the information with the specifications of the components used (hardware), as well as the line of programming (C++) added to article (line 151 to 161 and 469 to 476)
The EC sensor can measure between 0 and 20 mS cm-1. It supports a broad voltage input of 3–5 V, and its measurement accuracy is ±5%. The pH sensor can measure 0-14 pH, it supports a broad voltage input of 3~5 V, and its measurement accuracy is ±1%. The DO sensor can measure in the range of 0-20 mg L-1. It supports a broad voltage input of 3~5 V. Its measurement accuracy is ±2%. Meanwhile, the temperature sensor DS18B20 is a digital temperature sensor. The temperature range it can detect is -55 ~ 125 ℃, it has an inherent temperature resolution of 0.5 ℃, and it supports a broad voltage input of 3~5 V. All sensors, except for the temperature sensors, require maintenance and calibration every six months. There is also a GSM module in the model. This GSM module has Quad-band 850/900/1800/1900MHz and can connect to any global GSM network with any 2G SIM.
The invention can control the water parameters in the system with its inexpensive Arduino microcontroller, hardware, and software developed in the original open-source C++ software language, and thus is open to development. C++ is a widely used, general-purpose programming language. In 1983, the name was changed to C++. Since the invention is open-source, it can be modified by the user and integrated into the any-size system. The developed control kit can provide the parameters to reach the desired level by interfering with the liquid in the soilless agriculture system. Additionally, data can be monitored from smartphones. |
|
e) Cost of each kit component, including 3D printing added to materia and methods (line 171 to 172)
The total production cost of the kit, including all of the hardware and 3D printing costs, is approximately USD 500. |
|
f) Necessary explanations have been added. All the sections revised. |

Round 2
Reviewer 3 Report
Dear Author,
The author has responded to most of my comments, recommendations, and suggestions. In this step, I made some text box comments in the attached file.
The aspect I highlight is that the author must insert a description of the statistical analysis (Linear relationships) in the Material and Methods topic.
The manuscript may be accepted after meeting these adjustments.

Author Response
Response to Reviewer#3:
Suggested revisions by Reviewer#3 round_2
|
a) Describe each parameter again here, with the respective units (SI): Electrical conductivity (EC), power of hydrogen (pH), dissolved oxygen (DO), and temperature (T). |
|
b) Describe IoT applications |
|
c) Describe in the Material and Methods the statistical analysis |
Corrections given by Reviewer#3 round_2
|
a) Added (line 86 to 92). Electrical conductivity (EC) is the electrical resistivity of water, and SI unit symbol Siemens (S) is the unit of electric conductance (mS cm-1). The power of hydrogen or potential of hydrogen (pH) measures the acidity or alkalinity of water-soluble substances. Seven is neutral; lower values are more acidic and alkaline and have no units. Dissolved oxygen (DO) is the amount of oxygen in water; the SI unit symbol is mg L-1. Temperature (T) represents the water temperature, and the SI unit symbol is Kelvin (K) or Celsius (°C). |
|
b) Added (line 106 to 110) With the developing technology, smart devices, mobile phones, and thousands of devices connected to the internet have been developed. All technological or electronic devices (objects) produced are now addressable and usable in the network environment. IoT is a set of systems that control or regulate physical objects to see, perceive data, think, make decisions, share data, and communicate with each other. |
|
c) Added (line 334 to 339) During the study, the values obtained from the liquid in the system with the help of professional devices were compared with those obtained simultaneously from the developed kit. For this, the intraclass correlation coefficient (ICC) was found using a one-way random, single-measures method. ICC can be used when quantitative measurements are made on units organized into groups, and it describes how strongly units in the same group resemble each other. |
